# A zero-valent palladium cluster-organic framework

Xiyue Liu[1], James N. McPherson[1]✉, Carl Emil Andersen[1],
Mike S. B. Jørgensen[1], René Wugt Larsen[1], Nathan J. Yutronkie[2],
Fabrice Wilhelm[2], Andrei Rogalev[2], Mónica Giménez-Marqués[3],
Guillermo Mínguez Espallargas[3], Christian R. Göb[4] &
Kasper S. Pedersen[1]✉

Acquiring spatial control of nanoscopic metal clusters is central to their function as efficient multi-electron catalysts. However, dispersing metal clusters on surfaces or in porous hosts is accompanied by an intrinsic heterogeneity that hampers detailed understanding of the chemical structure and its relation to reactivities. Tethering pre-assembled molecular metal clusters into polymeric, crystalline 2D or 3D networks constitutes an unproven approach to realizing ordered arrays of chemically well-defined metal clusters. Herein, we report the facile synthesis of a {Pd$_3$} cluster-based organometallic framework from a molecular *triangulo*-Pd$_3$(CNXyl)$_6$ (Xyl = xylyl; **Pd$_3$**) cluster under chemically mild conditions. The formally zero-valent Pd$_3$ cluster readily engages in a complete ligand exchange when exposed to a similar, ditopic isocyanide ligand, resulting in polymerization into a 2D coordination network (**Pd$_3$-MOF**). The structure of **Pd$_3$-MOF** could be unambiguously determined by continuous rotation 3D electron diffraction (3D-ED) experiments to a resolution of ~1.0 Å (>99% completeness), showcasing the applicability of 3D-ED to nanocrystalline, organometallic polymers. **Pd$_3$-MOF** displays Pd$^0_3$ cluster nodes, which possess significant thermal and aerobic stability, and activity towards hydrogenation catalysis. Importantly, the realization of **Pd$_3$-MOF** paves the way for the exploitation of metal clusters as building blocks for rigidly interlocked metal nanoparticles at the molecular limit.

Ultrasmall metallic clusters, approaching the molecular level, receive tremendous attention for improved and atom-efficient catalysts[1–4]. Synthetic inorganic chemistry hosts the possibility to create well-defined and perfectly monodisperse metal *clusters* encapsulated by pendant ligand scaffolds, which by Cotton's original definition exhibit significant bonding between metal atoms (or ions)[5]. Such molecular entities are, however, not suited for applications in heterogeneous catalysis. Reticular chemistry, the synthetic art of metal-organic frameworks (MOFs), crafts bespoke networks from well-defined cationic inorganic nodes and anionic organic linkers[6–9]. Until very recently, these approaches have been restricted to structural building units containing metal ions in moderate to high oxidation states, which are a far cry from the active sites on metallic nanoparticle catalysts. Nonetheless, MOFs have been employed to immobilize clusters and nanoparticles within their well-defined pores, and such systems have outperformed state-of-the-art nanoparticle catalysts[10,11]. While

[1]Department of Chemistry, Technical University of Denmark, Kemitorvet 207, DK-2800 Kgs, Lyngby, Denmark. [2]European Synchrotron Radiation Facility (ESRF), CS 40220, 38043, Grenoble Cedex 9, France. [3]Instituto de Ciencia Molecular (ICMol), Universidad de Valencia, Paterna, 46980 Valencia, Spain. [4]Rigaku Europe SE, Hugenottenallee 167, 63263 Neu-Isenburg, Germany. ✉e-mail: jnemc@kemi.dtu.dk; kastp@kemi.dtu.dk

promising, such approaches compromise the inherent pore volume of the parent MOFs, and the clusters are not stabilized by the framework lattice to the same extent as atoms at the structural nodes, and so can degrade and leach from the material over time. More attractive would be to stabilize metallic clusters as the structural nodes in such crystalline frameworks, which until now has not been realized. Recently, a couple of zero- or low-valent MOF prototypes have been reported[12,13], and we outlined a strategy to incorporate zero-valent metal-carbonyl nodes in MOFs[14,15]. Indeed, low-valent metals are not only relevant in catalysis, for example, their unique affinities to gasses and coordinative flexibility have also been exploited to control the selectivity and pore volumes in porous MOF materials[16,17]. Polynuclear clusters are prevalent amongst metal carbonyls, which possess metal-nuclearities up to 145 atoms[18], and interesting catalytic properties[19,20], yet however,

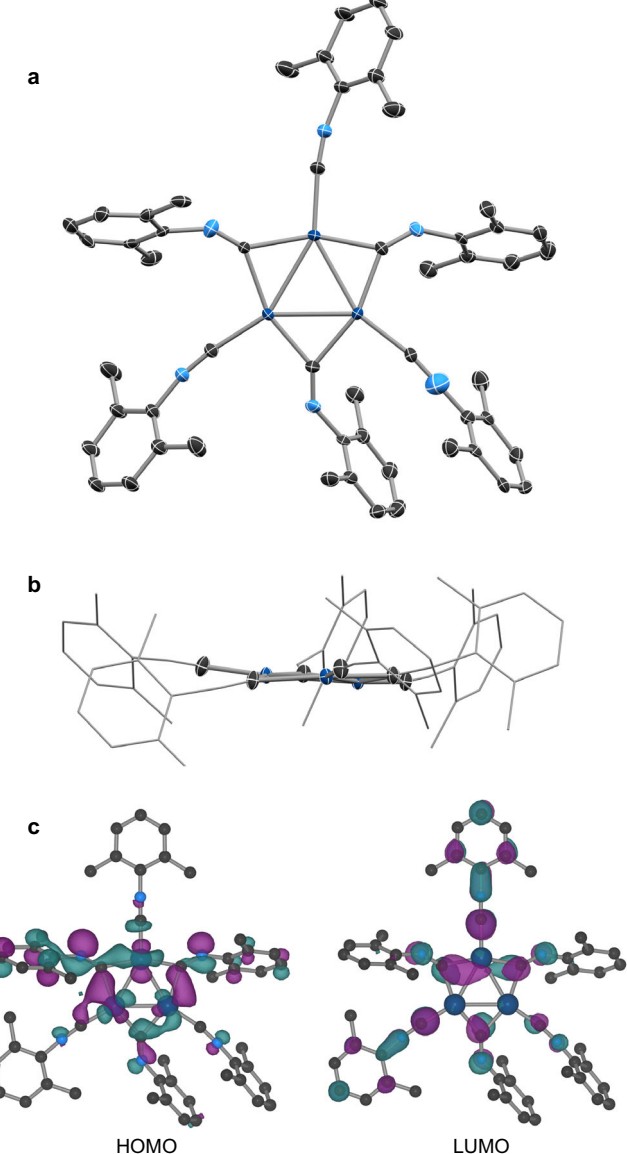

**Fig. 1 | Single-crystal X-ray diffraction and DFT results on Pd₃. a** SC-XRD structure of **Pd₃** (thermal ellipsoids drawn at 50% probability level, C (black), N (pale blue), Pd (dark blue)) shown perpendicular to the Pd₃ plane, while **b** shows a view along the plane (with C- and N-atoms outside the immediate Pd₃ coordination sphere represented by wireframe). H atoms and the second components of positional disorder are omitted for clarity. DFT-calculated highest occupied molecular orbitals (HOMO) and lowest unoccupied molecular orbitals (LUMO) for DFT geometry-optimized **Pd₃** (**c**; isosurface value = 0.03).

have not been utilized as structural framework nodes. While "paddle-wheel" complexes with metal-metal bonding have been employed extensively for the construction of framework materials[21], there are no examples of frameworks featuring formally zero-valent metal cluster nodes. Conceptually, the realization of such materials could be achieved by chemical substitution at preassembled, molecular clusters. Here, we present a two-dimensional cluster-based framework obtained by direct ligand exchange on an isocyanide-supported Pd(0) molecular cluster. Isocyanides (C≡N–R) are isolobal to CO, and so stabilize electron-rich metals while allowing synthetic and structural diversity. Others have exploited these possibilities to construct crystalline, polymeric materials featuring low- or zero-valent mononuclear nodes[22–24]. Molecular, triangular clusters supported by isocyanide ligands have been known for decades, and some systems characterized crystallographically[25–30]. Chemically well-defined Pd(0) triangles are conveniently synthesized from mononuclear, organometallic Pd(0) precursors and monotopic isocyanide ligands[26]. Analogous reactions employing ditopic isocyanide linkers, on the contrary, resulted in rapid, uncontrolled precipitation of intractable products[31]. Our strategy was instead to separate the {Pd₃} cluster formation and network propagation steps. This method exploits straightforward ligand exchange reactions to access crystalline framework architectures with triangular {Pd₃} cluster nodes.

## Results and discussion

### Synthesis, structure, and characterization of the molecular {Pd₃} cluster

The molecular {Pd₃} cluster, *triangulo*-Pd₃(CNXyl)₆ (**Pd₃**), was isolated from the previously reported reaction of Pd(dba)₂ (dba = dibenzylideneacetone) with excess 2,6-dimethylphenyl isocyanide (CNXyl)[26], and was structurally characterized by conventional single-crystal X-ray diffraction (SC-XRD; Fig. 1a, Supplementary Fig. 2, Supplementary Table 1). The {Pd₃} cluster in **Pd₃** is supported by a total of six isocyanide ligands, forming a virtually planar coordination environment (Fig. 1b). The three Pd atoms form an isosceles triangle which is bisected by a mirror plane (Pd1–Pd2 2.6641(6) Å, Pd1–Pd1' 2.7142(7) Å, Pd–Pd'–Pd" 59.377(9)°, 61.247(18)°). Three of the isocyanide ligands are bound in a monotopic terminal binding mode, while the other three are coordinated in a $\mu_2$-bridging mode. Despite the different binding modes, the C–N bond lengths vary only slightly (1.145(9)–1.172(15) Å for the terminal ligands cf. 1.160(13)–1.212(8) Å for the bridging ligands), and overlap with the C–N bond found in the structure of unbound CNXyl (1.160(4) Å)[32]. The C–N–Ar bond angles are more diverse: the terminal ligands are close to 180° (168.0(6)–175.1(11)°), while the bridging ligands are significantly bent (139.9(6)–146.9(4)°). The infrared (IR) absorption spectra illustrate the different binding modes (Supplementary Fig. 9) with $v$(NC) of the terminal ligands assigned at 2086 cm⁻¹, and $v$(NC) of the bridging ligands assigned to the strong, broad bands at lower energies (between 1700 and 1925 cm⁻¹) due to significant $4d \rightarrow \pi^*$ back-bonding, consistent with the observations for related triangular clusters[25,27,30].

The electronic structure of **Pd₃** was elucidated by density functional theory (DFT) calculations. Whilst elemental {Pd⁰₃} clusters of $C_{2v}$ and $D_{3h}$ symmetry are predicted to exhibit a paramagnetic ground state[33,34], DFT suggests a stabilization of the singlet state in **Pd₃** with a singlet–triplet gap exceeding 2 eV. The presence of a singlet ground state is confirmed by bulk magnetization measurements. The HOMO and LUMO orbitals of the DFT-geometry optimized structure of **Pd₃** are shown in Fig. 1c. The decomposition of the LUMO into localized, atomic orbitals shows a vanishing Pd character. In essence, the LUMO can be described as a bonding interaction between all six out-of-plane NC $\pi^*$ orbitals. This behavior parallels that found in the 44-electron Chini's Pt₃($\mu$-CO)₃(CO)₃²⁻ cluster and Figueroa's isocyanide-carbonyl analog Pt₃($\mu$-CO)₃(CNAr^Dipp2)₃²⁻ [29,35]. Hoffmann and coworkers have discussed the electronic structure of the hypothetical 42-electron

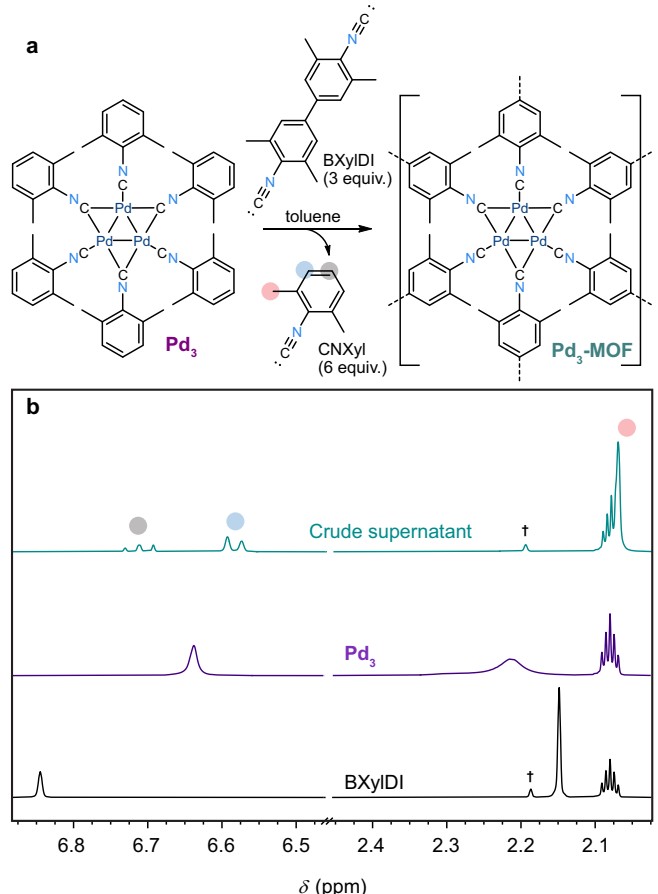

**Fig. 2 | Synthesis and $^1$H-NMR spectroscopy.** Synthesis of **Pd$_3$**-MOF (**a**) and $^1$H-NMR (400 MHz, toluene-$d_8$) spectra (**b**) of BXylDI (black trace), and **Pd$_3$** (purple trace) before and after mixing (green trace). Characteristic signals of unbound CNXyl in the spectrum of the crude supernatant are indicated with colored circles, while a trace impurity is indicated by the dagger, † Source data are provided as a Source Data file.

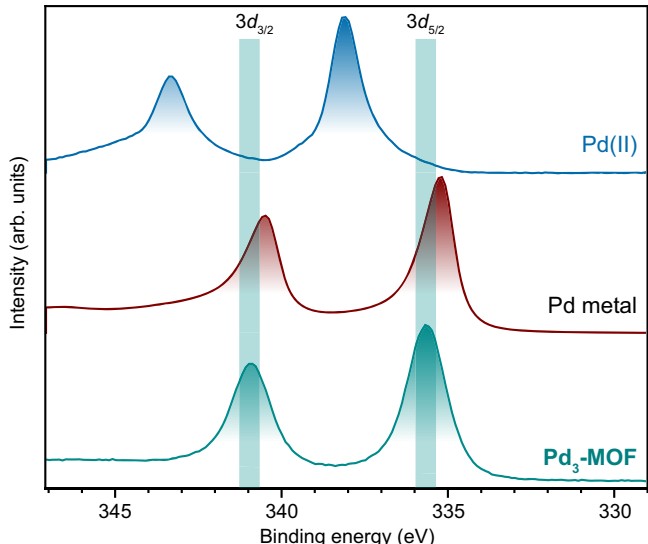

**Fig. 3 | X-ray photoelectron spectroscopy.** X-ray photoelectron spectra of **Pd$_3$**-MOF compared to metallic Pd (red trace) and a Pd(II) (K$_2$PdCl$_4$, blue trace) reference sample. The green panels mark the Pd $3d_{3/2}$ and $3d_{5/2}$ level binding energies for **Pd$_3$**-MOF. Source data are provided as a Source Data file.

Pt$_3$($\mu$-CO)$_3$(CO)$_3$ which is isoelectronic to **Pd$_3$**[36]. The a$_1$'($D_{3h}$)-symmetric HOMO orbital is lying in the plane of the Pt triangle and of predominant 5$d$ metal character (77%). For **Pd$_3$**, the HOMO shows significant, but diminished Pd character (18%), primarily through in-plane 4$d$ contributions (Fig. 1c), but carries significant admixture with the ligand scaffold orbitals. This behavior parallels that previously found in formally Pd(0)–isocyanide complexes[37].

## Synthesis, structure, and characterization of the {Pd$_3$} cluster framework

The fluxional $^1$H-NMR (400 MHz) spectrum of **Pd$_3$** in toluene-$d_8$ (purple trace in Fig. 2, Supplementary Fig. 6) suggests that the bridging and terminal ligands are rapidly exchanging, an effect also observed in Pt$_3$(CN–R)$_6$ analogs[25]. Mixing **Pd$_3$** and BXylDI (3 equiv.; BXylDI = 4,4'-diisocyano-3,3',5,5'-tetramethylbiphenyl) in toluene-$d_8$ results in the rapid precipitation of a bright red powder. After four hours, the complete replacement of the CNXyl ligands by BXylDI ligands was confirmed by $^1$H-NMR analysis of the supernatant, with no trace of **Pd$_3$** or BXylDI remaining in the solution phase (Fig. 2).

The elemental analysis of the powder product, **Pd$_3$**-MOF, was consistent with a formulation of Pd$_3$(BXylDI)$_3$, and **Pd$_3$**-MOF was stable under inert atmospheres up to at least 100 °C (see Supplementary Figs. 14 and 15). The zero-valence of Pd in **Pd$_3$**-MOF was confirmed by X-ray photoelectron spectroscopy (XPS, see Fig. 3). The Pd $3d_{3/2}$ and $3d_{5/2}$ level binding energies at 340.9 and 335.7 eV, respectively, for **Pd$_3$**-MOF, were in good agreement to those of a metallic Pd reference

sample measured under the same conditions (340.5 and 335.2 eV), and more than 2 eV lower than the binding energies of a Pd(II) reference (K$_2$PdCl$_4$, at 343.3 and 338.1 eV). The diffuse reflectance UV–vis spectrum of **Pd$_3$**-MOF suggests a similar electronic structure to that of **Pd$_3$**, whose solution UV–vis spectrum is dominated by charge transfer transitions (cf. Supplementary Fig. 7). Furthermore, the IR absorption spectrum of **Pd$_3$**-MOF features broad and intense NC vibrational modes as observed in **Pd$_3$** (Supplementary Fig. 9). Notably, no changes were observed in the IR absorption spectra of **Pd$_3$**-MOF after exposure to atmospheric air for 48 h (Supplementary Fig. 10), while **Pd$_3$** dissolved in toluene decomposed within two hours on exposure to air (Supplementary Fig. 8). **Pd$_3$**-MOF tolerated exposure to a range of solvents (H$_2$O, C$_6$H$_6$, THF, CH$_3$CN, CH$_2$Cl$_2$, $i$-PrOH) with no dramatic changes in the IR absorption spectra or powder X-ray diffractograms observed after soaking and subsequent drying (see Supplementary Figs. 16 and 17). **Pd$_3$**-MOF showed only minor uptake of N$_2$ at −196 °C (Supplementary Fig. 11a), from which a Brunauer–Emmett–Teller (BET) surface area of ~70 m$^2$ g$^{-1}$ was calculated. Gravimetric CO$_2$ gas sorption studies revealed a non-negligible accessible porosity (Supplementary Fig. 11b), with 1.0 mol of CO$_2$ adsorbed at 25 °C and 9 bar per mol of {Pd$_3$} cluster.

The powder X-ray diffractogram (Fig. 4) shows several, but only broad reflections suggesting the presence of nanoscopic crystalline domains in **Pd$_3$**-MOF. Indeed, scanning electron microscopy (SEM) images of **Pd$_3$**-MOF (Supplementary Fig. 12) show a homogeneous powder consisting of aggregates of small rods of μm length, but with significantly shorter other dimensions. Such small particle sizes make structural characterization by single-crystal X-ray diffraction impossible, even at synchrotron facilities. The structural characterization of **Pd$_3$**-MOF is therefore hampered by both the level of detail of the powder X-ray diffractogram and the size of the potential single crystals.

Structural information of **Pd$_3$**-MOF in lieu of a crystal structure may be obtained by Pd $K$-edge X-ray absorption near edge structure (XANES) and extended X-ray absorption fine structure (EXAFS) spectroscopy at either the Pd $K$- and $L_3$-edges (Fig. 5a, b). The XANES spectra of **Pd$_3$** and **Pd$_3$**-MOF are very similar, reflecting an essentially identical Pd electronic configuration (Fig. 5a). Further insights into the electronic structures were acquired from the comparison of the normalized $L_3$-XANES spectra of **Pd$_3$** and **Pd$_3$**-MOF to Pd reference samples with established formal oxidation states (Fig. 5b). At first glance, the molecular Pd(0) and the metallic Pd references, share a common, weak resonance at the

edge. This contrasts with the Pd(II) reference, which exhibits a stronger resonance (double in intensity) originating from $2p \rightarrow 4d$ transitions. Moreover, the maxima of the resonances in **Pd₃** and **Pd₃-MOF** are found at the same photon energies as the Pd(0) molecular reference. On the basis of these observations, and in agreement with the XPS results, we can assign a formal Pd(0) oxidation state in both **Pd₃** and **Pd₃-MOF**. Furthermore, the geometric similarity between **Pd₃** and **Pd₃-MOF** was evidenced by almost the same Fourier-transformed (FT) EXAFS spectra (Fig. 5c), confirming the conservation of the local atomic arrangement of {Pd₃} clusters during the synthesis of **Pd₃-MOF**.

Definitive structural elucidation was pursued by continuous rotation 3D electron diffraction (3D-ED) analysis. Transmission electron microscopy pictures and electron diffraction images reveal submicron single crystals in **Pd₃-MOF** (Supplementary Fig. 3). By merging diffraction data of several crystals, the 3D-ED crystal structure of **Pd₃-MOF** could be determined (Fig. 6) to a resolution of 1.0 Å (99.4% completeness, $R_1 = 17.9\%$).

The structure reveals a two-dimensional polymeric framework consisting of interlocked triangular {Pd₃} nodes. The {Pd₃} nodes

remain supported by six isocyanide ligands, with three isocyanide groups bridging neighboring Pd atoms and the other three bound terminally to one Pd atom each (Fig. 6). While all terminally bound isocyanides have nearly linear C–N–Ar angles (171–174°), two of the bridging isocyanides are heavily bent (130–134°). One of the three bridging isocyanides exhibits a C–N–Ar angle similar to that of the terminally bound isocyanides (171°), likely induced by structural strain enforced from the framework. The three Pd–Pd′ distances (2.665(5)–2.773(5) Å) are within 3σ, but are otherwise consistent with those of **Pd₃**. The 2D sheets of **Pd₃-MOF** are separated by 4.1 Å and are offset (Supplementary Fig. 4), such that the π-stacked ligands sit in the obvious place: directly above and below the rhomboid 'voids'. Simulation of the EXAFS spectrum using the local coordination environment for a single Pd center of the 3D-ED structure of **Pd₃-MOF** (Fig. 5c) shows good agreement with the experimental data. Finally, the comparison of the simulation of the powder X-ray diffractogram using the 3D-ED structure with the experimental data yields a reasonable match and suggests the complete absence of crystalline impurity phases.

To demonstrate the conceptual resemblance of **Pd₃-MOF** to immobilized and supported Pd nanoparticles, we performed preliminary studies of the catalytic hydrogenation of styrene, which is a well-established activity of both nanoparticulate and bulk, metallic Pd (see Supplementary Note 10)[38–43]. While the activity of **Pd₃-MOF** was modest compared to some other supported Pd nanoparticles (5% conversion after 4 h at 5 wt% catalyst loading; see Supplementary Fig. 18 and Supplementary Table 6), there is considerable scope to optimize the reaction conditions to improve substrate access to Pd⁰₃ sites and mass transport kinetics during the heterogeneous reaction. No conversion was observed in the absence of **Pd₃-MOF**, while the molecular **Pd₃** decomposed upon exposure to H₂, without any consumption of styrene (Supplementary Fig. 19). No evidence of decomposition of **Pd₃-MOF** (e.g., dissolved organic linker) was observed in the ¹H NMR spectra of the reaction mixtures.

In conclusion, we have presented the first example of a metal-organic framework constructed on metallic cluster nodes. In essence, **Pd₃-MOF** is a realization of spatially confined, perfectly monodisperse metal clusters rigidly embedded in solid support, which remains stable above 100 °C and tolerates exposure to air and a wide range of solvents (including water). Initial reactivity studies reveal apparent parallelism

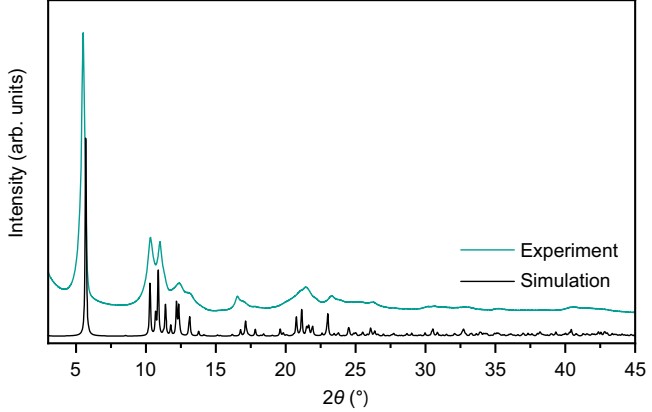

**Fig. 4 | Powder X-ray diffraction.** Room-temperature powder X-ray diffractogram (Cu Kα, λ = 1.5406 Å) of **Pd₃-MOF** (green trace) with a simulated diffractogram (black trace) calculated from the 3D-ED crystal structure (vide infra). Source data are provided as a Source Data file.

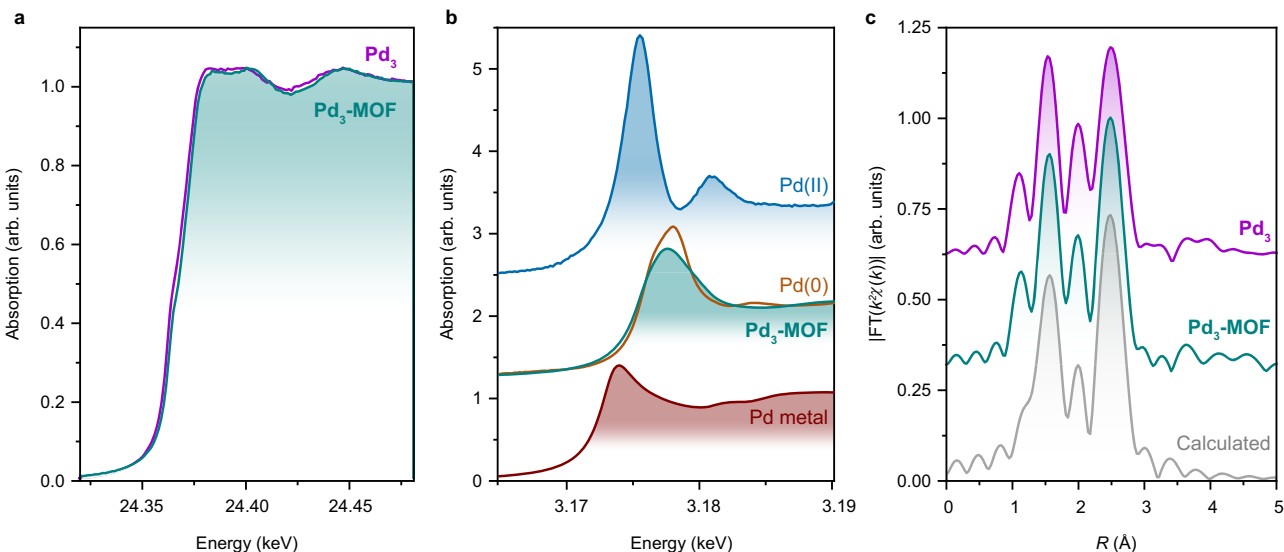

**Fig. 5 | X-ray spectroscopy. a** Pd K-edge XANES spectra of **Pd₃** (purple trace) and **Pd₃-MOF** (green trace). **b** Pd L₃-edge XANES spectra of **Pd₃-MOF**, Pd foil (red trace), Pd(0) (Pd(PPh₃)₄) (orange trace), and Pd(II) (PdCl₂(NCPh)₂) (blue trace) reference samples. **c** R-space Pd K-edge FT-EXAFS spectra of **Pd₃** and **Pd₃-MOF** together with a simulation from the 3D-ED structure (gray trace) of **Pd₃-MOF**. Source data are provided as a Source Data file.

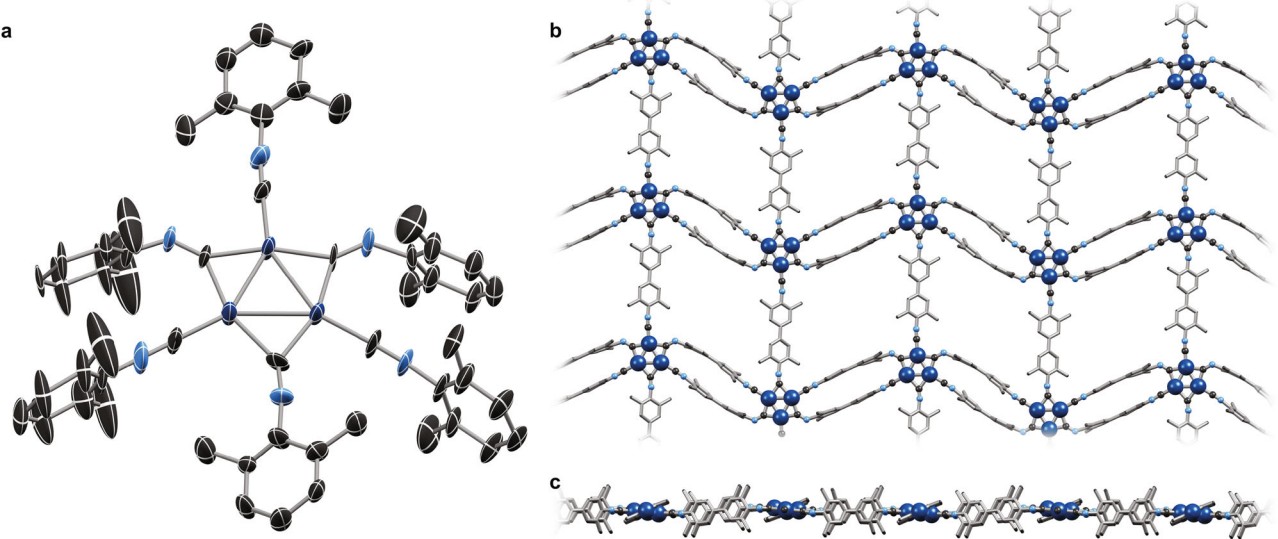

**Fig. 6 | 3D-ED structure of Pd₃-MOF.** Single-crystal 3D-ED structure of **Pd₃-MOF** showing the asymmetric unit (50% thermal ellipsoids, C (black), N (pale blue), Pd (dark blue); **a**) and views of an extended two-dimensional sheet perpendicular (**b**) and parallel (**c**) to the mean plane through the sheet. H atoms are omitted for clarity. Selected bond lengths and angles: Pd–Pd: 2.66–2.73 Å; Pd–C$_{terminal}$: 1.75–2.01 Å; Pd–C$_{bridging}$: 2.02–2.20 Å; N–C$_{terminal}$: 1.15–1.23 Å; N–C$_{bridging}$: 1.18–1.25 Å; Pd–Pd′–Pd″: 59–62°; C$_{terminal}$–N–Ar: 172–173°; C$_{bridging}$–N–Ar: 130, 134, 170°.

between the {Pd⁰₃} cluster nodes in **Pd₃-MOF** and Pd nanoparticle catalysts. The two-dimensional nature of **Pd₃-MOF** sheets may be amenable to delamination, which would increase the accessibility of the embedded {Pd⁰₃} clusters. Whilst intensively studied, structural characterization of such individual sheets remains difficult or impossible[44–46]. The modular synthetic approach developed herein could be easily expanded to realize truly porous, three-dimensional analogs of **Pd₃-MOF**, by using, for example, tri- or tetra-topic isocyanide linkers. Importantly, the advent of 3D electron diffraction as an upcoming routine technique in the structure determination of MOFs alleviates the intrinsic obstacles associated with the crystallization of such zero-valent MOF materials. We envisage that a plethora of new cluster-based MOFs with potential porosity is now accessible through chemical substitution at isolable molecular, zero-valent cluster building blocks.

## Methods

### General methods

Unless otherwise stated, all synthetic manipulations were performed inside a dry, argon-filled glove box, and commercial reagents were used as supplied.

**Synthesis of BXylDI.** The reported synthesis[47] was adapted as follows under ambient atmosphere. 3,3′,5,5′-Tetramethylbenzidine (2.214 g, 9.21 mmol) and CHCl₃ (3.2 mL, 40 mmol) were dissolved in CH₂Cl₂ (100 mL) and added to aqueous KOH (45%, 100 mL). Benzyltriethylammonium chloride (12 mg, 0.05 mmol) was added, and the two-phase reaction mixture was stirred at reflux for 10 h. The mixture was then diluted with water (100 mL), and the organic phase was separated after the mixture had cooled to ambient temperature. The aqueous phase was further extracted with CH₂Cl₂ (3 × 100 mL), and all organic phases were then combined and washed with water (3 × 100 mL) and dried over Na₂SO₄. The organic solvents were removed, and off-white BXylDI (892 mg, 37%) was recovered after two successive recrystallizations from hot iso-propanol. ¹H NMR (chloroform-$d$, 400 MHz, 298 K) $\delta$ 7.27 (s, 1H, H$_{aryl}$), 2.48 (s, 3H, CH₃) ppm. FTIR (NC) $\nu_{max}$: 2122 (vs) cm⁻¹.

**Synthesis of Pd₃.** The reported synthesis[26] was adapted as follows. Pd(dba)₂ (dba = dibenzylideneacetone, 290 mg, 0.505 mmol) was added in portions to a pale yellow solution of CNXyl (165 mg, 1.26 mmol) in hexane (2 mL). The reaction mixture was left to stir for 20 min at room temperature. The suspended solids in the mixture changed color from black to red, then gray, and finally settled on a pale yellow. Hexane was removed in vacuo, and diethyl ether (18 mL) was added followed by stirring for another 45 min. The pale red solution was decanted away, and the red solid residue was filtered and washed with diethyl ether (3 × 2 mL). The product was dried under a dynamic vacuum for 20 min to afford Pd₃(CNXyl)₆ (**Pd₃**, 163.2 mg, 98%) as a fine red powder. ¹H NMR (toluene-$d_8$, 400 MHz, 298 K) $\delta$ 6.63 (brs, 18H, H$_{aryl}$), 2.28 (vbrs, 6H, CH₃), 2.21 (vbrs, 30H, CH₃) ppm. UV–vis (toluene; $\lambda_{max}$ ($\varepsilon_{max}$): 297 (4.3 × 10⁴), 334 (4.5 × 10⁴), 414 (3.0 × 10⁴), 443 (3.8 × 10⁴), 545 nm (3.2 × 10³ L cm⁻¹ mol⁻¹). FTIR (NC) $\nu_{max}$: 2086 (vs), 1922 (br), 1815 (m), 1765 (m) cm⁻¹. Analysis calculated for Pd₃(C₉H₉N)₆ C: 58.63, H: 4.92, N: 7.60, Pd: 28.86%. Found (standard error in the final significant figure across duplicates given in parentheses) C: 58.13(2), H: 4.97(1), N: 7.50(1), Pd: 28.53(1)%.

**Synthesis of Pd₃-MOF.** A pale-yellow solution of BXylDI (28.7 mg, 0.110 mmol) in toluene (6 mL) was filtered through a nylon membrane (0.22 µm pores) and added to a red solution of **Pd₃** (39.7 mg, 0.036 mmol) in toluene (4 mL). The mixture was stirred for two days at room temperature. The pale orange liquid phase was decanted away, and the bright red residues were suspended in toluene (2 mL) and then collected by filtration. The residues were washed with toluene (3 × 1.5 mL) and then dried under a dynamic vacuum for 2 h to afford Pd₃(BXylDI)₃ (**Pd₃-MOF**, 32.2 mg, 81%) as a bright red powder. Slow diffusion by carefully layering the BxylDI solution onto the **Pd₃** solution did not improve the crystallinity of **Pd₃-MOF**. FTIR (NC) $\nu_{max}$: 2086 (vs), 1991 (m), 1727 (br) cm⁻¹. Analysis calculated for Pd₃(C₁₈H₁₆N₂)₃ C: 58.95, H: 4.40, N: 7.64, Pd: 29.02%. Found (standard error in the final significant figure across duplicates given in parentheses) C: 58.56(1), H: 4.36(1), N: 7.58(1), Pd: 28.94(1)%.

### Single-crystal X-ray diffraction

Single crystals of BXylDI and **Pd₃** were suspended in polybutenes and mounted on a loop onto a Supernova DualSource diffractometer. Experiments were conducted in a nitrogen stream at 120 K using Cu-K$\alpha$ radiation ($\lambda$ = 1.5406 Å). The structures were solved with ShelXT[48] using intrinsic phasing and refined by least squares with ShelXL[49]

using Olex2[50]. Further refinement details specific to each structure follow.

**BXylDI.** Colorless and transparent single crystals of BXylDI, suitable for single-crystal X-ray diffraction, were grown as a super-concentrated solution of BXylDI in boiling THF cooled to room temperature. The structure (Supplementary Fig. 1) was solved in the orthorhombic *Fddd* space group, with a quarter of the BXylDI molecule defined in the asymmetric unit. See Supplementary Tables 1 and 4 for crystallographic data and key metric data, respectively. The two xylyl rings are not coplanar, with an angle of 51.01(6)° between the normals to the mean plane through each 6-membered aromatic ring. The two isocyanide donors at either end of the BXylDI molecule are separated by 12.129(3) Å.

**$Pd_3$.** Red microcrystals of $Pd_3$ were dissolved in a minimum of toluene under argon. The solution was cooled to −22 °C and after a couple of days, red crystals suitable for X-ray diffraction were obtained. Key crystallographic and metric data are summarized in Supplementary Tables 1 and 4, respectively. The structure of $Pd_3$ was solved in the monoclinic *Cm* space group. A rigid body restraint (RIGU) was applied to keep anisotropic displacement parameters stable during refinement. Two of the six CNXyl ligands were disordered across the (020) mirror plane which bisects the molecule. As such, two classes of ligand residues were assigned: "TERM" for terminal $\mu_1$-CNXyl, and "BRID" for bridging $\mu_2$-CNXyl. SAME restraints were then applied to all TERM residues to locate the positions of atoms in the disordered terminal CNXyl ligand. This procedure was also followed for the "BRID" residues to locate the atomic positions of the disordered bridging CNXyl. Once the positions for all C and N atoms in the six CNXyl ligands were found, the atoms were modeled with anisotropic displacement parameters, and H atoms were included using HFIX. Atoms were, where possible, moved off special positions but then freely refined. Despite this, the model was most stable when Pd02, C1_4 and N1_4 were modeled on the mirror plane. Once stable, the SAME restraints for the TERM residues were removed, and the model was refined until stable. This was repeated for BRID residues. The (020) mirror plane bisects $Pd_3$ (Supplementary Fig. 2), and so only half the molecule is defined in the asymmetric unit. Two ligands are positionally disordered across the mirror plane: the terminal CNXyl bound to Pd02 (residue 2), and the CNXyl which bridges the two crystallographically equivalent Pd01 atoms (residue 4).

### Continuous rotation 3D electron diffraction
In an argon-filled glovebox, $Pd_3$-MOF powders were ground gently and transferred into 4 mL sintered glass vials for transport to Rigaku. On arrival, a lacey carbon grid was then placed directly into the vial, the vial was shaken, and the grid was transferred quickly into the sample holder and the electron diffractometer. Continuous rotation 3D electron diffraction data were acquired using the dedicated electron diffractometer Rigaku XtaLAB Synergy-ED, equipped with a HyPix-ED detector by Rigaku Oxford Diffraction[51]. Data acquisition was performed at ambient temperature with an electron wavelength of 0.0251 Å (200 kV). The data were processed using CrysAlisPro[52], the structure was solved using ShelXT[48] and subsequently refined using the kinematical approximation with olex.refine in the crystallographic program suite Olex2[50]. By merging the data of five individual crystals, a completeness of 99.4% up to a resolution of 1.00 Å was achieved for $Pd_3$-MOF. The crystals screened for the diffraction analysis can be seen in Supplementary Fig. 3. Key crystallographic and metric data are summarized in Supplementary Tables 2–4, respectively.

### Powder X-ray diffraction
Inside an argon-filled glove box, samples of $Pd_3$-MOF were ground into fine powders, and transferred into glass capillaries ($\varnothing$ 0.7 mm, 0.01 mm wall thickness). Diffractograms were collected on a Malvern Panalytical Empyrean powder X-ray diffractometer, equipped with a

1Der detector, and using Cu K$\alpha$ ($\lambda$ = 1.5406 Å) radiation in capillary spinner mode at $V$ = 45 kV and $I$ = 40 mA. The samples were measured at room temperature between $3 \leq 2\theta \leq 80°$ with step size 0.008° and scan rate 0.01° s$^{-1}$. Powder X-ray diffraction on $Pd_3$ (Supplementary Fig. 5) was measured in reflection mode at room temperature between $3.5 \leq 2\theta \leq 60°$ with a step size of 0.008° and a scan rate of 0.015° s$^{-1}$.

### NMR spectroscopy
$^1$H NMR spectra were collected on a Bruker AVANCE 400 MHz system equipped with a 5 mm Prodigy CryoProbe. Spectra were processed using MestReNova[53] and calibrated against residual protic solvent signals[54]. The $^1$H NMR (400 MHz, toluene-$d_8$, 298 K) spectrum of $Pd_3$ is compared to that of a solution of unbound CNXyl in Supplementary Fig. 6. Despite the two different CNXyl binding modes-terminal and bridging-in $Pd_3$, only one singlet (black region in Supplementary Fig. 6) in the aromatic region of the $^1$H-NMR spectrum was observed, indicating rapid exchange (on the NMR timescale) between the ligands.

**Ligand exchange experiment.** Stock solutions of $Pd_3$ (19.9 mg, 18.0 μmol) in toluene-$d_8$ (2.1639 g), and BXylDI (14.1 mg, 54.2 μmol) in toluene-$d_8$ (2.1637 g) were prepared. The solution of BXylDI (25 μmol g$^{-1}$) was added to the solution of $Pd_3$(CNXyl)$_6$ (8.2 μmol g$^{-1}$), and the mixture was initially shaken by hand. A dark red solid immediately precipitated from the mixture. After standing for four hours, an aliquot of the pale orange supernatant was collected and analyzed by $^1$H NMR spectroscopy.

### UV−vis absorption and diffuse reflectance spectroscopy
UV−vis and diffuse reflectance spectroscopy were measured on an Agilent Cary 5000 UV−Vis-NIR spectrophotometer. The UV−vis spectrum of $Pd_3$ (0.043 mM in toluene) was recorded in a 1 cm quartz cuvette, and the background was corrected by subtracting a spectrum of the neat solvent. Diffuse reflectance spectroscopy was measured in a Praying Mantis cell.

### Infrared spectroscopy
Attenuated total reflectance (ATR) Fourier transform infrared (FTIR) spectra were obtained by a VERTEX80v FTIR vacuum spectrometer from Bruker Optics, GmbH. The FTIR instrument was equipped with a single reflection germanium ATR accessory (PIKE Technologies Inc.), a Ge/KBr beam splitter, and a liquid nitrogen-cooled MCT (HgCdTe) detector in combination with an air-cooled SiC thermal radiation source. The ATR spectra (2 cm$^{-1}$ resolution) were corrected for minor water vapor absorption, baseline-corrected, and ATR corrections were applied to compensate for the wavelength-dependent penetration depth of the probe beam.

### Gas sorption analysis
Low-pressure $N_2$ gas volumetric sorption isotherm was obtained using a TRISTAR II Plus apparatus (Micrometrics), at −196 °C. Gravimetric $CO_2$ gas sorption isotherms were recorded up to 9 bar in an IGA-100 gas sorption analyzer (Hidden Isochema), at 25 and 10 °C. Gravimetric equilibrium conditions corresponded to 600 s interval, and 0.001 mg min$^{-1}$ tolerance. A powder sample of ca. 40 mg $Pd_3$-MOF was activated for two hours at 100 °C under a dynamic vacuum immediately before each measurement. $Pd_3$-MOF integrity was confirmed by powder X-ray diffraction at the conclusion of all cycles.

### Scanning electron microscopy
Scanning electron microscopy images were measured on an AFEG 250 Analytical ESEM and generated with secondary electrons detected by Everhart−Thornley detector at the following settings: high tension = 10 kV, spot size = 3.5, working distance = 10.0 mm. $Pd_3$-MOF was ground, sprinkled on double-sided adhesive, conductive carbon tape, and coated with 4 nm Au prior to imaging.

## X-ray absorption spectroscopy

**Data acquisition.** At the ID12 beamline of the ESRF in Grenoble, X-ray absorption (XANES and EXAFS) of **Pd₃** and **Pd₃-MOF** were measured at both the Pd $K$- and $L_3$-edges, while reference samples of Pd foil, Pd(PPh₃)₄ and PdCl₂(NCPh)₂ were only measured at the Pd $L_3$-edge. For the $L_3$-edge experiments, the fundamental harmonics of Helios-II type undulator were used in a circular polarization mode. The incident beam was monochromatized using the <111> reflection of a pair of cryogenically cooled Si crystals, which provided an energy bandwidth of 0.44 eV. Higher-order harmonics were suppressed by a pair of B₄C mirrors installed downstream with respect to the monochromator. For the present $K$-edge experiments, the fifth harmonics of an APPLE-II type undulator were used in a linear polarization mode. During the EXAFS scan exploiting the third harmonics of the Si<111> mono-chromator, the undulator was permanently tuned to the maximum of its emission peak, in order to optimize the photon flux at the sample. All spectra were recorded in a total fluorescence yield detection mode using a Si photodiode.

**EXAFS modeling.** The EXAFS modeling was performed using the *Artemis*[55] software package. The paths used for simulating the EXAFS spectrum were calculated by FEFF[56]. Scattering amplitudes, phase shifts, photoelectron mean free paths, and path degeneracies were calculated by FEFF. The fitted parameters were thus the core–hole relaxation ($S_0^2$), inner potential correction ($\Delta E_0$), change in photo-electron mean free path ($\Delta R$), and its mean square displacement ($\sigma^2$). The best fit was obtained from paths generated for Pd01 (see Supplementary Table 5). The fitting procedure used data in the range of 2–12 Å$^{-1}$ in $k$-space while optimizing the fit in $R$-space in the range of 1–3 Å. We have applied a very simple fitting model, where we assumed the same parameter value for all paths. This holds true for $S_0^2$ and $\Delta E_0$, while it is a rough approximation of $\Delta R$ and $\sigma^2$. None-theless, we believe the resulting fit is sufficient to confirm that there are no significant changes in the coordination sphere of Pd in the measured sample compared to the 3D-ED model. The optimized parameters were $S_0^2 = 0.63(7)$, $\Delta E_0 = 9(1)$ eV, $\Delta R = 0.015(7)$ Å, $\sigma^2 = 0.0030(8)$ Å. A figure comparing the full $k$-space of **Pd₃-MOF** and **Pd₃** is provided in the Supplementary Information (Supplementary Fig. 13).

## X-ray photoelectron spectroscopy

XPS measurements were carried out on a Thermofisher Scientific Nexsa instrument, utilizing a monochromated Al Kα source at 1486.6 eV. The spectra were recorded with a 400 μm spot size and 50 eV pass energy. Charge neutralization was ensured by the flood gun providing electrons and Ar ions with low kinetic energies to the surface of the samples.

## Density functional theory calculations

All DFT calculations were performed using the ORCA 5 software package[57,58]. All energies were converged to $10^{-6}$ atomic units using the identity approximation[59] to accelerate the calculations. Geometry optimization converged effortlessly at the PBE0/def2-SVP level of theory[60–62] utilizing Grimme's D4 methodology to correct for London dispersion forces[63,64]. Single-point energy calculations were achieved using the TPSSh/def2-TZVP methodology[62,65] together with the NBO7 program to obtain natural population analyses[66,67]. In all cases, the effective core potential (ECP) approximation was used to account for the 28 inner core electrons on Pd atoms[68] as well as Weigend's def2/J auxiliary basis[69].

## Stability and reactivity studies

**Thermogravimetric analysis.** In an argon-filled glovebox, samples of **Pd₃-MOF** were ground into fine powders. Vials were charged with these **Pd₃-MOF** powders (15–20 mg) and analyzed on Mettler Toledo a TGA/DSC 1 STAR System equipped with a Mettler Toledo Gas Controller GC 100. The samples were heated at 2 °C min$^{-1}$ from 25 to 500 °C, under either nitrogen or ambient air atmospheres.

**Thermal stability of crystalline phase.** Capillaries (0.5 mm diameter, 0.01 mm wall thickness) were charged with **Pd₃-MOF** in an argon-filled glovebox, and the powder X-ray diffraction was measured at room temperature between $3 \leq 2\theta \leq 57°$ with 0.008° steps at 0.01° s$^{-1}$. The capillaries were then heated in an oven to either 100 or 140 °C for two hours and then remeasured using the same experimental parameters.

**Solvent stability.** In an argon-filled glovebox, samples of **Pd₃-MOF** were ground into fine powders. Glass vials were charged with **Pd₃-MOF** powders (~10 mg), and ~5 mL of either benzene, tetrahydrofuran, acetonitrile or dichloromethane were added, and these samples were left to soak for 6 h. Meanwhile, two vials charged only with **Pd₃-MOF** were cycled out of the glovebox and these **Pd₃-MOF** samples were soaked in ~5 mL of water or isopropanol under an ambient atmo-sphere. All solvents were then decanted off each sample, and the **Pd₃-MOF** powders were all dried under a dynamic vacuum or flow of N₂. The dried **Pd₃-MOF** samples were then analyzed by ATR-FTIR and powder X-ray diffraction ($3 \leq 2\theta \leq 50°$ with 0.008° step size and at 0.026° s$^{-1}$ scan rate in 0.5 mm diameter, 0.01 mm wall thickness capillaries), as described above.

**Air stability of Pd₃.** The decomposition of a toluene solution of **Pd₃** in air was monitored by UV–vis absorption spectroscopy as follows (Supplementary Fig. 8). In an argon-filled glovebox, **Pd₃** (ca. 10 mg) was dissolved in 20.0 mL of toluene. An aliquot of the solution (1.0 mL) was then diluted with toluene (10.0 mL, total volume = 11.0 mL) to obtain a concentration of ca. 45 μM. A 1 cm quartz cuvette was filled, sealed, and then removed from the glovebox and placed immediately in an Agilent Cary 5000 UV–Vis–NIR spectrophotometer. Spectra were recorded at 2 min intervals over two hours, and the Teflon cap was removed from the cuvette immediately after the third spectrum was acquired.

**Hydrogenation of styrene screening experiments.** The method reported by Fan et al.[39] was adapted as follows. In an argon-filled glo-vebox, a stock solution of styrene (330 mm) and mesitylene (54 mm) in toluene-$d_8$ was prepared, and 0.5 mL aliquots (0.165 mmol of styrene) were transferred into J. Young NMR tubes which were empty or charged with either **Pd₃-MOF** (0.9 or 1.6 mg, 0.8 or 1.5 μmol of "Pd₃", respectively) or **Pd₃** (1.2 mg, 1.1 μmol, 0.7 mol%). These were sealed, removed from the glovebox, and analyzed immediately by $^1$H NMR spectroscopy. The samples were then frozen inside the NMR tubes, which were then evacuated, and the tubes were then charged with H₂ (1 bar) after the solutions were thawed. The samples were then vigor-ously stirred for 1 min and analyzed again by $^1$H NMR spectroscopy before they were further immersed in an ultrasound bath (at 40 °C). The J. Young NMR tubes were transferred directly into a Bruker AVANCE 400 MHz NMR Spectrometer equipped with a 5 mm SmartProbe BB(F)-H-D after the time intervals indicated in Supple-mentary Figs. 18 and 19. Yields were calculated using the mesitylene standard. The reactions with **Pd₃-MOF** were performed in parallel with either the blank or **Pd₃** controls and halted either on the consumption of H₂ ($\delta$ 4.51) or on the decomposition of **Pd₃** complex (loss of signals at $\delta$ 6.65 and 2.22 and observation of non-coordinated CNXyl at $\delta$ 1.88).

## Data availability

All data are available in the main text or in the Supplementary Infor-mation and in the source data files. Source data are provided in this paper. The XAS data are also available from DOI: 10.15151/ESRF-ES-1176719512. Crystallographic information has been deposited in the Cambridge Crystallographic Data Centre under the accession codes CCDC 2293023 (BXylDI), 2293024 (**Pd₃**), and 2293025 (**Pd₃-MOF**). Source data are provided with this paper.

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

## Acknowledgements

K.S.P. thanks the Carlsberg Foundation for a Carlsberg Foundation Young Researcher Fellowship (CF21-0416), and the Independent Research Fund Denmark for a DFF-Research Project 1 grant (0135-00291B) and a Sapere Aude: DFF-Starting Grant (0165-00073B). We thank Jonas Michael-Lindhard (National Centre for Nanofabrication and Characterization Process Engineering, DTU Nanolab) for assistance with XPS measurements.

## Author contributions

K.S.P. and J.N.M. conceived and designed the research project. X.L. and J.N.M. performed the syntheses and NMR measurements. J.N.M. performed the X-ray crystallographic analyses and C.R.G. performed the 3D-ED crystallographic analyses. M.S.B.J. performed the DFT calculations. R.W.L. measured and obtained the IR data. A.R. and N.J.Y. performed and analyzed the X-ray absorption spectroscopic measurements. C.E.A., A.R. and F.W. modeled the EXAFS data. C.E.A. performed the UV–vis absorption and diffuse reflectance spectroscopy, and measured and obtained the SEM data. M.G.-M. and G.M.E. carried out the gas sorption experiments and performed the interpretation. The manuscript was written through the contributions of all authors.

## Competing interests

The authors declare no competing interests.
