## [Peer Review File · Nature Communications]

REVIEWER COMMENTS

Reviewer #1 (Remarks to the Author):

The manuscript by McPherson, Pedersen, and co-workers presents an interesting and novel perspective on isolating low-valent Pd (0) centres within crystalline matrices. In this work the low-valent Pd centres are incorporated as Pd₃ clusters which comprise the nodes of the framework. This article is well-written, and I would like to congratulate the authors of this interesting contribution. The paper focuses on pre-synthesising a labile molecular triangulo-Pd₃ complex which is subsequently exposed to a ditopic isocyanide ligand, resulting in spontaneous polymerisation into a 2D coordination polymer. This approach is novel and distinguished from strategies so far employed to form low-valent MOFs. As noted by the authors, there is a paucity of low-valent MOFs in the literature and the novel synthesis strategy described in this contribution can likely be extended to other low-valent clusters. The species formed in this research are challenging to synthesise and characterise, and the authors present excellent work which provides insight into the formed material. This reviewer finds the ideas expressed in the article clear and sufficiently novel for publication in *Nature Communications*. Considering the quality and presentation of the work by the authors, I suggest only minor revisions, which are listed in the points below:

1. The authors titled the manuscript "A palladium cluster-organic framework"; however, this title is somewhat vague and does not provide much context for the content of the paper. In particular, it does not emphasise the remarkable low-valent state of the cluster nodes. Hence, this reviewer kindly suggests including the word 'low valent' or 'zero valent' or otherwise presenting the oxidation state of the Pd centres in the title.
2. As the authors clearly state, low-valence metal complexes are electron-rich species of special interest for catalytic applications. However, these species are also subject to interest in other fields, such as adsorption (*J. Am. Chem. Soc.* 2023, 145, 37, 20492–20502, *Chem. Sci.*, 2021, 12, 14893–14900). This reviewer suggests that the authors comment on the possible application of their approach in this emerging application for MOFs with accessible low-oxidation state metals.
3. The authors comment that Pd₃-MOF exhibits no changes to the IR spectrum when exposed to atmospheric air for 48 hr. Presumably, the point of this comment is to highlight that isolation of the low-valent Pd₃ species within a crystalline matrix prevents decomposition upon exposure to air; however, the authors do not report any analysis of the material's porosity. Is the material expected to be porous? It would be helpful to clarify the nature of the 'stabilisation' of Pd₃ within the framework; for example, are the Pd₃ moieties stabilised by their site-isolation within the framework (presumably prohibiting agglomeration into larger particles or disproportionation) or because they are actually inaccessible to O₂ molecules due to incorporation within a non-porous framework.
4. One of the most important features of MOFs is their porosity. However, the authors did not report any adsorption studies in this manuscript. This reviewer imagines that the material is either not stable upon activation or, owing to the staggered arrangement of 2D sheets, is not accessible to gas molecules. This reviewer suggests that the authors comment on these aspects, the accessibility of the low-valent Pd centres within the framework would appear to be an important consideration in any application, although the reviewer understands that this study presents a 'proof of concept' for the novel synthesis approach using low-valent clusters. The 2D nature of the material would suggest that in-situ exfoliation of the sheets would render the Pd centres accessible to substrates in catalytic applications if the current assembly does not permit access.
5. The authors state that PXRD patterns obtained from Pd₃-MOF show good agreement with the simulated pattern suggesting 'the complete absence of impurity phases'. Considering the reliance on diffraction techniques implied in this assessment, perhaps a more accurate statement would read 'the complete absence of crystalline impurity phases'.
6. Did the authors attempt to grow crystals with tritopic or tetratopic isocyanide linkers? It is conceivable that tri or tetratopic linkers could form three-dimensional frameworks which impart

greater site isolation to the Pd₃ clusters.

Reviewer #2 (Remarks to the Author):

The present work reports, indeed, an original strategy to prepare a MOF based on the use of Pd(0)₃ nanoclusters as preformed SBUs. This new strategy may lead to novel families of MOFs with application in different interesting fields like catalysis. Therefore I think it deserves publication in Nat. Comms after addressing some points.

- Pd oxidation state should be further ensured by carrying out XPS measurements and/or, alternatively, measuring IR under CO.

- I miss gas adsorption experiments to verify if this material presents a certain permanent porosity (even if modest).

- Overall, the material (as well as the synthetic strategy) is very interesting but I would have liked that the authors gave any relevant property for this material (measuring their catalytic behaviour)

- What about the stability of the MOF ? (solvents, Temperature, etc). This point should be further discussed.

Reviewer #3 (Remarks to the Author):

The paper by Pedersen and co-workers extended the MOF chemistry about "A palladium cluster-organic framework". Key to success is the use of ligand-exchange approach (known synthetic method in MOFs) to prepare the Pd₃-cluster MOF, in which the Pd₃-cluster nodes (reported in 1983) monodisperse in the rigid solid support. This work is interesting in the MOF synthetic field, although I don't feel that the synthetic strategy represents a big advance. In addition, the authors did not show any properties or applications of Pd₃-MOF. Therefore, I cannot recommend this manuscript to be published in Nature Commun.

1. "In conclusion, we have presented the first example of a metal-organic framework constructed on metallic cluster nodes." This statement might be wrong.

2. The more evidence and data are needed to prove the valence of Pd.

3. The characterizations (BET, TGA, etc.) generally associated with MOFs are not sufficiently observed.

4. One of the main driving forces for the synthesis of uniformly dispersed materials with metal atoms is their high catalytic activity, how about Pd₃-MOF? Is that stable and active enough to promote certain organic transformations in a heterogeneous way?

RESPONSE TO REVIEWERS' COMMENTS

Dear referees,

We are delighted to read the overall highly positive and highly constructive comments, questions, and criticism raised by the expert referees. We have addressed all the points raised in full below and in the attached revised manuscript and Supplementary Information files. We are thankful for the insightful comments, which we find have facilitated a significant improvement of our manuscript, both in terms of further corroboration of the scientific conclusions, as well as the readability.

We hope that the referees find that our manuscript is now acceptable for publication in Nature Communications.

Yours sincerely,

Kasper S. Pedersen, James N. McPherson, Xiyue Liu, on behalf of all authors.

REVIEWER COMMENTS

Reviewer #1 (Remarks to the Author):

The manuscript by McPherson, Pedersen, and co-workers presents an interesting and novel perspective on isolating low-valent Pd (0) centres within crystalline matrices. In this work the low-valent Pd centres are incorporated as Pd₃ clusters which comprise the nodes of the framework. This article is well-written, and I would like to congratulate the authors of this interesting contribution. The paper focuses on pre-synthesising a labile molecular triangulo-Pd₃ complex which is subsequently exposed to a ditopic isocyanide ligand, resulting in spontaneous polymerisation into a 2D coordination polymer. This approach is novel and distinguished from strategies so-far employed to form low-valent MOFs. As noted by the authors, there is a paucity of low-valent MOFs in the literature and the novel synthesis strategy described in this contribution can likely be extended to other low-valent clusters. The species formed in this research are challenging to synthesise and characterise, and the authors present excellent work which provides insight into the formed material. This reviewer finds the ideas expressed in the article clear and sufficiently novel for publication in Nature Communications.

Reply: We are flattered to read that the referee finds our manuscript represents “an interesting and novel perspective” and “find the ideas expressed in the article clear and sufficiently novel for publication in Nature Communications”.

Considering the quality and presentation of the work by the authors, I suggest only minor revisions, which are listed in the points below:

1. The authors titled the manuscript “A palladium cluster-organic framework”; however, this title is somewhat vague and does not provide much context for the content of the paper. In particular, it does not emphasise the remarkable low-valent state of the cluster nodes. Hence,

this reviewer kindly suggests including the word 'low valent' or 'zero valent' or otherwise presenting the oxidation state of the Pd centres in the title.

Reply: We agree with the referee that it is relevant to emphasize the valence state of the cluster in the title. We have modified the title slightly to "A zero-valent palladium cluster-organic framework".

2. As the authors clearly state, low-valence metal complexes are electron-rich species of special interest for catalytic applications. However, these species are also subject to interest in other fields, such as adsorption (J. Am. Chem. Soc. 2023, 145, 37, 20492–20502, Chem. Sci., 2021, 12, 14893-14900). This reviewer suggests that the authors comment on the possible application of their approach in this emerging application for MOFs with accessible low-oxidation state metals.

Reply: We fully agree with the referee that the discussion should be broadened to capture the interest of such materials in other fields. We have amended the discussion.

3. The authors comment that Pd₃-MOF exhibits no changes to the IR spectrum when exposed to atmospheric air for 48 hr. Presumably, the point of this comment is to highlight that isolation of the low-valent Pd₃ species within a crystalline matrix prevents decomposition upon exposure to air; however, the authors do not report any analysis of the material's porosity. Is the material expected to be porous? It would be helpful to clarify the nature of the 'stabilisation' of Pd₃ within the framework; for example, are the Pd₃ moieties stabilised by their site-isolation within the framework (presumably prohibiting agglomeration into larger particles or disproportionation) or because they are actually inaccessible to O₂ molecules due to incorporation within a non-porous framework.

Reply: We have provided additional experiments to investigate the stability of Pd₃-MOF in response to this and the other reviewers' comments. We agree with the referee that it could be expected that the air-stability of Pd₃-MOF results from the inaccessibility of the Pd₃ moieties to O₂ in the non-porous material. To elucidate this, we have performed additional experiments to explore the stability and porosity of Pd₃-MOF, which we have included in the revised manuscript and Supplementary Information file.

Most remarkably, the X-ray photoelectron spectrum of Pd₃-MOF shows no evidence of even trace amounts of palladium oxidation in the surface layers of the nanocrystals. We have provided the XPS spectrum in Fig. 3 of the revised manuscript. For comparison, we provide below the XPS spectrum of a moderately air-sensitive palladium(0), which was handled under identical conditions to Pd₃-MOF, which showed comparable amounts of Pd(II) and Pd(0) in the surface layers. This result, combined with the now-included reactivity studies and non-negligible porosity (at least to CO₂) suggest that the apparent stability of Pd₃-MOF is not entirely governed by inaccessibility of the Pd₃ clusters.

4. One of the most important features of MOFs is their porosity. However, the authors did not report any adsorption studies in this manuscript. This reviewer imagines that the material is either not stable upon activation or, owing to the staggered arrangement of 2D sheets, is not accessible to gas molecules. This reviewer suggests that the authors comment on these aspects, the accessibility of the low-valent Pd centres within the framework would appear to be an important consideration in any application, although the reviewer understands that this study presents a 'proof of concept' for the novel synthesis approach using low-valent clusters. The 2D nature of the material would suggest that in-situ exfoliation of the sheets would render the Pd centres accessible to substrates in catalytic applications if the current assembly does not permit access.

*Reply: We thank the reviewer for these relevant comments. Indeed, gas-sorption experiments were not originally included, as the crystal structure did not reveal any significant internal surface area. We have now performed these experiments and included them in our revised manuscript. Interestingly, while N_2 was not able to access any internal porosity, up to ~ 1 mol of CO_2 per mol of Pd_3 cluster was adsorbed by **Pd_3 -MOF** at $25^\circ C$ and 9 bar. As correctly mentioned by the referee, "this study represents a 'proof of concept' for the novel synthesis approach using low-valent clusters." Exfoliation could indeed increase accessibility of the 2D sheets to gas molecules, although the methodology to obtain chemically well-defined 2D coordination sheets is far from trivial. We have added a discussion on these aspects to the concluding remarks in our revised manuscript.*

5. The authors state that PXRD patterns obtained from Pd_3 -MOF show good agreement with the simulated pattern suggesting 'the complete absence of impurity phases'. Considering the reliance on diffraction techniques implied in this assessment, perhaps a more accurate statement would read 'the complete absence of crystalline impurity phases'.

Reply: We fully agree with the referee and have revised the quoted sentence.

6. Did the authors attempt to grow crystals with tritropic or tetratopic isocyanide linkers? It is conceivable that tri or tetratopic linkers could form three-dimensional frameworks which impart greater site isolation to the Pd₃ clusters.

Reply: We thank the referee for the suggestion that indeed is an interesting further elaboration of our methodology, which we will pursue in our laboratory. We have included a brief comment in the concluding section where we discuss such strategies to increase the dimensionality and porosity of such materials.

Reviewer #2 (Remarks to the Author):

The present work reports, indeed, an original strategy to prepare a MOF based on the use of Pd(0)₃ nanoclusters as preformed SBUs. This new strategy may lead to novel families of MOFs with application in different interesting fields like catalysis. Therefore I think it deserves publication in Nat. Comms after addressing some points.

Reply: We are delighted to read that the referee finds our work “original” and that they recommend publication upon addressing the points below. We have provided a detailed reply to all criticism raised and hope that the referee finds that the manuscript now can be accepted for publication in Nature Communications.

- Pd oxidation state should be further ensured by carrying out XPS measurements and/or, alternatively, measuring IR under CO.

*Reply: We fully agree with the referee, and the requested XPS measurements are now included in the revised manuscript (Fig. 3). Notably, the experimentally observed Pd 3d_{3/2, 5/2} binding energies in **Pd₃-MOF** agree well with those measured in metallic Pd, rather than in Pd(II). Furthermore, we have now performed additional L-edge XANES measurements in order to compare **Pd₃-MOF** with other formally zero-valent palladium materials. We have expanded our discussion to incorporate these new results, and provided the new data in Fig. 5. Our new XPS and XANES data both unambiguously suggest the assignment of zero-valency in **Pd₃-MOF**, which has now been clarified in the main text. We are uncertain what additional information might be gained from measuring IR under CO, but can perform this experiment if deemed necessary by the referee.*

- I miss gas adsorption experiments to verify if this material presents a certain permanent porosity (even if modest).

Reply: As mentioned above in response to similar comments by reviewer #1, we have now undertaken these experiments, and included the (expected) results in the revised manuscript.

- Overall, the material (as well as the synthetic strategy) is very interesting but I would have liked that the authors gave any relevant property for this material (measuring their catalytic behaviour)

Reply: We are delighted that the referee finds our synthetic strategy and material to be “very interesting”. Indeed, the main purpose of this manuscript is to communicate a novel chemical

route and the existence of cluster-based frameworks—unlocked by cutting-edge electron crystallography, which are of interest across a diverse range of applications, including catalysis. We have now diversified the discussion of this interest in the main text. Furthermore, we have performed a proof-of-principle reactivity study that demonstrates that the Pd⁰₃ clusters within **Pd₃-MOF** possess similar activity to Pd nanoparticles, specifically, as catalysts for the hydrogenation of styrene. These new data and a discussion of the resemblance to bulk Pd have been introduced in the revised manuscript.

- What about the stability of the MOF ? (solvents, Temperature, etc). This point should be further discussed.

Reply: Neither the IR spectrum, nor the powder X-ray diffractograms of **Pd₃-MOF** change when exposed to a wide range of common solvents (water, isopropanol, benzene, tetrahydrofuran, acetonitrile, and dichloromethane). We have added a discussion to the main text, and the new data are shown in Figs S16 and S17. In addition, we have studied the thermal stability by thermogravimetric analysis and powder X-ray diffractometry of heated samples. These experiments demonstrate a significant thermal robustness of **Pd₃-MOF** extending above 100 deg. C. and these new data are provided and discussed in the revised manuscript.

Reviewer #3 (Remarks to the Author):

The paper by Pedersen and co-workers extended the MOF chemistry about “A palladium cluster-organic framework”. Key to success is the use of ligand-exchange approach (known synthetic method in MOFs) to prepare the Pd₃-cluster MOF, in which the Pd₃-cluster nodes (reported in 1983) monodisperse in the rigid solid support. This work is interesting in the MOF synthetic field, although I don't feel that the synthetic strategy represents a big advance. In addition, the authors did not show any properties or applications of Pd₃-MOF. Therefore, I cannot recommend this manuscript to be published in Nature Commun.

Reply: We fully agree with the referee that ligand exchange and the synthesis of molecular clusters, in itself, do not “represent a big advance”. However, we would like to clarify, and have done so in the revised manuscript, that our synthesis and structural characterization approach goes a significant step further. We agree with the referee that the synthetic approach is not overly complicated, but we view the simplicity of the approach as a strength and something that will be of broad appeal to the interdisciplinary readership of Nature Communications. It is furthermore our position that our study is timely, given the paucity of low- or zero-valent MOFs (as emphasized by referees #1 and #2) and the current explosion of interest in electron crystallography. Cluster-nodes in MOFs, or more generally, in coordination polymers, are unprecedented. The obstacles of crystallization of such materials have hitherto prevented their discovery. We have now demonstrated that facile chemical substitution at preassembled molecular clusters yield nanocrystalline specimens allowing for atomic resolution crystal structures only with contemporary electron diffraction. In addition to underlining these aspects, we have performed further studies that demonstrate the stability and reactivity of **Pd₃-MOF** akin to Pd nanoparticles.

We hope that our arguments and the improved clarity of the manuscript text may change the mind of this referee.

1. “In conclusion, we have presented the first example of a metal-organic framework constructed on metallic cluster nodes.” This statement might be wrong.

Reply: *We have naturally surveyed the literature comprehensively, and are confident that no examples of MOFs or coordination polymers, more generally, with well-defined, mono-disperse zero-valent cluster nodes have been reported.*

2. The more evidence and data are needed to prove the valence of Pd.

Reply: *We have now included both Pd X-ray photoelectron spectra (Fig. 3) and Pd L₃ X-ray absorption near-edge spectra (XANES, Fig. 5b) for Pd₃-MOF and reference samples with well-established valence.*

As also discussed in response to the same query from referee #2, the observed Pd 3d_{3/2,5/2} binding energies in Pd₃-MOF agree best with those of metallic Pd, rather than in the Pd(II) reference. We have expanded our discussion to incorporate these new results. In conclusion, our new XPS and XANES data both unambiguously suggest the assignment of zero-valency in Pd₃-MOF, which has now been clarified in the main text.

3. The characterizations (BET, TGA, etc.) generally associated with MOFs are not sufficiently observed.

Reply: *We have performed gas sorption (N₂ at 77 K and CO₂ at 283 and 298 K) and thermogravimetric analysis of Pd₃-MOF, and the results from these are now discussed in the main text. The 70 m² per gram of Pd₃-MOF BET surface area calculated from the N₂ isotherm at 77 K is in rough agreement with a back-of-the-envelope calculation of the surface area of 1 gram of 100 x 100 x 100 nm cubes (40 m²), suggesting negligible internal porosity that is accessible by N₂ molecules. However, CO₂ molecules, with a smaller kinetic diameter (3.30 vs 3.64 Å respectively for CO₂ and N₂), appear to be able to access narrow pores within Pd₃-MOF, with up to ~1.1 mol of CO₂ adsorbed per mol of {Pd₃} at 283 K and 9 bar. The TGA data (acquired both under N₂ and atmospheric air; Fig. S14) and powder X-ray diffractograms of heated samples (Fig. S15) demonstrate significant thermal stability of Pd₃-MOF, which has now been described in the main text.*

4. One of the main driving forces for the synthesis of uniformly dispersed materials with metal atoms is their high catalytic activity, how about Pd₃-MOF? Is that stable and active enough to promote certain organic transformations in a heterogeneous way?

Reply: *We agree with the referee that comments on the possible chemical reactivity of Pd₃-MOF is relevant, albeit not the focus of the manuscript. We have now demonstrated that Pd₃-MOF is “stable and active enough to promote certain organic transformations in a heterogeneous way”. As also mentioned above in response to a similar query from referee #2, we have now included experimental data on the catalytic hydrogenation of styrene—a benchmark reaction for Pd nanoparticle catalysts. We have provided a discussion on these results and a comparison to previously reported Pd systems.*

REVIEWERS' COMMENTS

Reviewer #1 (Remarks to the Author):

I would suggest the manuscript be published in its current form. The authors have addressed all my comments.

Reviewer #2 (Remarks to the Author):

I think that the authors have taken seriously my concerns (and dare to say that laso those of other referees). I am more than pleased with changes made and thus I suggest to accept the manuscript as it is.

Reviewer #3 (Remarks to the Author):

Now the manuscript can be published in current form.

RESPONSE TO REVIEWERS' COMMENTS

Dear referees,

We would like to thank you for your diligence and suggestions, which have considerably improved our manuscript. We are most grateful. We are delighted that the manuscript will now be published in Nature Communications, and are very appreciative of the time that was spent by all involved.

Yours sincerely,

Kasper S. Pedersen, James N. McPherson, Xiyue Liu, on behalf of all authors.

REVIEWER COMMENTS

Reviewer #1 (Remarks to the Author):

I would suggest the manuscript be published in its current form. The authors have addressed all my comments.

Reply: Thank you for your enthusiasm and helpful comments and suggestions.

Reviewer #2 (Remarks to the Author):

I think that the authors have taken seriously my concerns (and dare to say that laso those of other referees). I am more than pleased with changes made and thus I suggest to accept the manuscript as it is.

Reply: Thank you for your valuable suggestions and support of this work.

Reviewer #3 (Remarks to the Author):

Now the manuscript can be published in current form.

Reply: Thank you for your consideration. We are very pleased that we were able to address your concerns in our revised manuscript.